# IL-1β knockdown inhibits cigarette smoke extract-induced inflammation and apoptosis in vascular smooth muscle cells

**Hongfeng Jiang**[1]*, **Zhangqiang Guo**[2], **Kun Zeng**[1], **Haiyan Tang**[1], **Hanxuan Tan**[1], **Rui Min**[1], **Caihua Huang**[1]

1 Department of Geriatrics, Wuhan Fourth Hospital, Affiliated Puai Hospital of Tongji Medical College of Huazhong University of Science and Technology, Wuhan, China, 2 Department of Emergency Medicine, Wuhan Fourth Hospital, Affiliated Puai Hospital of Tongji Medical College of Huazhong University of Science and Technology, Wuhan, China

* jhf2@sina.com

## Abstract

### Objective

This study was aimed to investigate the role of interleukin-1β (IL-1β) in cigarette smoke extract (CSE)-induced apoptosis in vascular smooth muscle cells and the underlying mechanism in a rat derived cell line.

### Methods

Rat thoracic aortic smooth muscle cells (A7r5) were divided into six groups including control, CSE (model), CSE+ overexpression empty vector (OvExp-EV), CSE+IL-1β knockdown (KD), and CSE+ IL-1β knockdown empty vector (KD-EV). The mRNA expression levels of IL-1β and pregnancy-associated plasma protein A (PAPP-A) were detected by quantitative polymerase chain reaction (qPCR). The apoptosis of A7r5 cells was detected by flow cytometry. The expression levels of inflammatory mediators (TNFα, IL-6 and IL-8) and apoptotic proteins (Bax and Bcl-2) were determined by western blot.

### Results

CSE induced significant apoptosis in vascular smooth muscle cells ($P < 0.01$) and elevated the mRNA levels of IL-1β and PAPP-A ($P < 0.01$). CSE administration increased protein expression of Bax, TNF-α, IL-6, and IL-8, with significantly reduced Bcl-2 expression ($P < 0.01$). IL-1β knockdown significantly decreased cell apoptosis via regulating the expression of these proteins ($P < 0.05$ or $P < 0.01$).

### Conclusion

IL-1β is involved in CSE-induced PAPP-A expression and apoptosis in vascular smooth muscle cells, which might be considered as a target for preventing of cardiovascular diseases caused by cigarette smoking.

**Data Availability Statement:** All relevant data are within the paper and its Supporting Information files.

**Funding:** This study was supported by Wuhan Municipal Health Commission Project WX19D63.

The funders had no role in study design, data collection and analysis, decision to publish, or preparation of the manuscript.

**Competing interests:** The authors have declared that no competing interests exist.

**Abbreviations:** IL-1β, interleukin-1β; IL, interleukin; CSE, cigarette smoke extract; OvExp, overexpression; EV, empty vector; KD, knockdown; qPCR, quantitative polymerase chain reaction; PAPP-A, plasma protein A; IGFBP-4, insulin-like growth factor-binding protein-4; IGF-1, insulin-like growth factor-1; DMEM, Dulbecco's Modified Eagle Medium; TNF-α, tumor necrosis factor-α; GAPDH, glyceraldehyde-3-phosphate dehydrogenase; ICAM1, adhesion molecules intercellular adhesion molecule-1; VCAM1, vascular cell adhesion molecule-1.

## 1. Introduction

The idea that smoking is unhealthy is profoundly ingrained in the minds of general population. Accumulated evidence on the smoking-related risks has also been widely reported. However, the burden of smoking-induced diseases continues to rise, becoming one of the most critical concerns affecting the global healthcare system [1]. In 2019, the number of smokers increased to 1.1 billion worldwide, with 7.7 million deaths caused by the associated diseases [2].

Smoking is a vital risk factor for cardiovascular diseases, especially the coronary artery diseases [3]. The younger the smoker, the greater the risk of cardiovascular diseases, whereas smoking cessation reduces the risks [4]. Smoking leads to several complications such as endothelial impairment, thrombosis, inflammation, abnormal lipid metabolism, atherosclerosis formation and induces rupture of arterial plaques [5]. It has also been shown [6] that vascular smooth muscle phenotypic transformation, apoptosis, and calcification are closely related to the development of atherosclerosis.

The cytotoxicity test of the cigarette smoke is commonly conducted using the cigarette smoke condensate and cigarette smoke extract (CSE) [7]. Previous study [8] showed the activation of major atherosclerotic key parameters by CSE, such as adhesion molecules intercellular adhesion molecule-1 (ICAM1), vascular cell adhesion molecule-1(VCAM1), selectin E.

Animal studies have suggested that Interleukin-1β (IL-1β) is an important cytokine, whose secretion and activation promotes the formation of atherosclerosis and the instability of vascular plaque [9–11]. The roles of IL-1β in CSE-induced inflammation and apoptosis in vascular smooth muscle cells are rarely known.

Pregnancy-associated plasma protein A (PAPP-A) expresses in macrophages, vascular endothelial and smooth muscle cells, which is a metalloproteinase for insulin-like growth factor (IGF) system. After release into the bloodstream, PAPP-A cleaves insulin-like growth factor-binding protein-4 (IGFBP-4), resulting in increased release of free insulin-like growth factor-1 (IGF-1) and elevated biological activity. IGF-1 binds to its receptors in the smooth muscle cells and subsequently triggers a series of signaling cascades, promoting the development of atherosclerosis and increasing plaque vulnerability. PAPP-A serves as a useful predictive biomarker for vulnerable plaques, and plays a vital role in the early identification, risk stratification, and prognostic evaluation of acute coronary syndromes [12, 13].

Smoking increases the expression level of circulating inflammatory mediators and pro-inflammatory cytokines (eg. TNF-α, IL-1, IL-6, and IL-8), which are potent stimulators for PAPP-A expression. In addition, smoking also decreases the production of anti-inflammatory interleukin-10 (IL-10) [14–16]. Previous researches reported that pro-inflammatory cytokines TNF-α and IL-1β induced PAPP-A expression in a dose-dependent manner in cultured human fibroblasts and vascular smooth muscle cells [17, 18]. However, whether PAPP-A involves in the process of CSE-induced inflammation and apoptosis in vascular smooth muscle cells is rarely reported.

We speculate that the expression intensity of IL-1β might be closely related to the progression of atherosclerosis caused by smoking. In this study, gene knockout and overexpression were performed by molecular technology to interfere with IL-1β in rat thoracic aortic smooth muscle cells. The expression level of IL-1β and other cytokines in smooth muscle cells by CSE intervention was detected. By analyzing the effect of IL-1β gene knockout on apoptosis and expression of other cytokines of smooth muscle cells interfered by CSE, we can find new targets for preventing the harm of smoking to cardiovascular diseases.

## 2. Materials and methods

### 2.1. Regents and antibodies

Rat A7r5 cells were obtained from the Chinese academy of sciences cell bank. Dulbecco's Modified Eagle Medium (DMEM) was purchased from TBD Co., Ltd. ELISA kits were obtained from Wuhan Elisa Lab Technology Co., Ltd. Endonucleases *Nhe* I, *Kpn* I, *Age* I and *EcoR* I were purchased from NEB Co., Ltd. Opti-MEM was purchased from Gibco Co., Ltd. Transfection reagents were purchased from Beijing Chreagen Biotechnology Co., Ltd. q-PCR reagents were purchased from KAPA Biosystems Co., Ltd. Bcl-2 associated X protein (Bax) antibody, tumor necrosis factor-α (TNF-α) antibody, interleukin-6 (IL-6) antibody and IL-8 antibody were purchased from Bioswamp Co., Ltd.

### 2.2 Preparation of CSE and screening of concentrations

Smoke from cigarettes (commercially available Wuhan brand cigarettes with a tar content of 15 mg/stick and a smoke nicotine content of 1.2 mg/stick) was collected using a gas collection tube Balinzhave and dissolved in 10 mL of serum-free DMEM (Tianjin TBD Co. Ltd.; Cat#TBD10569). DMEM (Procell Life Science & Technology Co., Ltd.; Cat#164210–500) containing 10% FBS (Procell Life Science & Technology Co., Ltd.; Cat#PM150210) was adjusted to pH = 7.4 and then filtered with a 0.22 μm pore size filter membrane to collect the CSE stock solution. The stock solution was then diluted at the concentration of 2.5%, 5%, 10%, 20%, and 40%, stored for subsequent experiments. Rat A7r5 cells were resuscitated and cultured in DMEM at 37˚C in an incubator supplemented with 5% $CO_2$. The cells were divided into 6 groups. CSE solutions (0%, 2.5%, 5%, 10%, 20% and 40%) were added to each group. After CSC treatment, the culture medium were centrifuged and the supernatants were collected. An ELISA assay was performed for selecting the optimal concentration of CSE for further experiments.

### 2.3 ELISA

The cells in each group were centrifuged (400xg) for 20 min. The supernatant was collected. The standards were diluted according to the instructions of the PAPP-A ELISA kit. The samples were added into the wells on ELISA plate and incubated at room temperature for 30 min. After washing, the chromogenic agent was added into each well. The reaction was stopped by adding the stopping solution at room temperature and incubated avoiding light for 10 min. The absorbance of each well was measured at 450 nm ($OD_{450}$). The regression equation of the standard curve was calculated based on the concentration of the standard and the OD value. Finally, the concentration of PAPP-A was determined.

### 2.4 Plasmids

For IL-1β knockdown, synthetic oligonucleotides containing the rat IL-1β splice variant RNA interference target gene was synthesized, annealed, and ligated into the pLKO.1-EGFP vector. For IL-1β overexpression, synthetic primers were designed to obtain the target gene. Double digestion was used to digest the vector and the target gene. The vector and the target gene were ligated and then transferred into the competent cells. The recombinant plasmids containing positive clones were screened. The expression efficiency was verified by transfection into the smooth muscle cells. The expression level of the transfected genes was detected after 48 h of incubation.

## 2.5 Grouping and treatment

Cells ($5 \times 10^5$ cells per well) were divided into 6 groups, control, CSE+IL-1β (model), CSE+ overexpression empty vector (OvExp EV), CSE+IL-1β knockdown (KD), and CSE+ IL-1β knockdown empty vector (KD EV). These plasmids were transfected into smooth muscle cells. Except for the control group, all groups were administrated with 40% CSE and were incubated for 8 hours, and then cells were harvested for further investigation.

## 2.6 Real time fluorescent quantitative PCR

For the detection of IL-1β and PAPP-A mRNA expression, total RNA in the harvested cells ($1 \times 10^6$ cells) was homogenized in triazole. The homogenate was centrifuged for 1 min ($13,000 \times g$, 4°C). The RNA concentration was measured by NanoDrop ND-2000 Spectrophotometer (NanoDrop Technologies, Wilmington, DE), and cDNA was synthesized using a total of 1 μg RNA using QuantiTect® Quantiscript reverse-transcriptase and RT Primer Mix (Qiagen), according to the manufacturer's protocol. cDNA was synthesized by reverse transcription, and then q-PCR amplification was performed using it as a template. The reaction procedure was: denaturation at 95°C for 5 s, annealing at 56°C for 10 s, and extension at 72°C for 25 s, for a total of 40 cycles. IL-1β, PAPP-A, and glyceraldehyde-3-phosphate dehydrogenase (GAPDH) were detected in the cells. After the reaction, the respective amplification and melting curves were analyzed. PCR primers (Table 1) were synthesized by Wuhan Tianyi Huayu Gene Technology Co., Ltd.

## 2.7 Analysis of cell apoptosis by flow cytometry

A suspension containing $1 \times 10^6$ cells was centrifuged at 400 g for 5 min at 4°C. The supernatant was discarded, and 200 μl of pre-cooled PBS was added to resuspend the cells. 5 μl of Annexin V-PE and 7-AAD were added, mixed gently, and incubated for 30 min. Then, 300 μl of PBS was added, and eventually, the samples were analyzed using flow cytometry (BD Accuri™ C6 Plus Flow Cytometer, Auckland, NZ).

## 2.8 Western blotting

Cells were collected and lysed. Total proteins were extracted. After protein quantification by BCA Kit, 20 μg of protein was separated by SDS-PAGE electrophoresis. The proteins were transferred to the membrane and blocked in 5% skim milk before being probed overnight at 4°C with the primary antibodies. Primary antibodies such as rabbit Bax (Bioswamp Wuhan Beiyinle Biotechnology, China) (Cat#PAB32583, 1:1000), rabbit Bcl-2 (Bioswamp Wuhan Beiyinle Biotechnology, China) (Cat#PAB30599, 1:1000), rabbit GAPDH (Bioswamp Wuhan Beiyinle Biotechnology, China) (Cat#PAB36269, 1:1000) and goat IgG (Bioswamp Wuhan Beiyinle Biotechnology, China) (Cat#SAB43714, 1:10000), were used. After washing the

**Table 1. Primer sequences.**

| Primer | Sequence | Fragment size(bp) |
|---|---|---|
| IL-1β-F | CAAGCAACGACAAAATCCC | 147 |
| IL-1β-R | CAAACCGCTTTTCCATCTTC | |
| PAPP-A-F | CACTTGGGCGGTATTGTC | 108 |
| PAPP-A-R | ACGGAAGATGTGATAGAGGC | |
| GAPDH-F | CAAGTTCAACGGCACAG | 138 |
| GAPDH-R | CCAGTAGACTCCACGACAT | |

membrane, a secondary antibody was added and incubated with the membrane for 1 hour at room temperature. The membrane was washed, and analyzed by a chemiluminescence analyzer (Tanon-5200 Shanghai Tianneng, China) using a chemiluminescence reagent (Beyo ECL Plus (Beyotime, China). Finally, the band intensity was determined by using TANON Gell Imaging System Software.

## 2.9 Statistical analysis

Statistical analysis was performed using Prism software (version 8.02; GraphPad Software, San Diego, CA, USA). One-way ANOVA was performed for comparison between multiple groups. For post hoc test, Holm-Sidak was used to compare between two groups. $P < 0.05$ was considered as statistically significant.

## 3. Results

### 3.1 Screening the concentrations of CSE intervention

We found that the expression level of PAPP-A was increased exponentially compared to the control group in a dose-dependent manner ($P < 0.01$, Fig 1). We detected the concentration of PAPP-A as 547, 593, 804, 922, 1069 and 1395 pg/ml when 0%, 2.5%, 5%, 10%, 20%, and 40% CSE were administrated. Compared with the group without CSE intervention, administration of 40% CSE showed the highest PAPP-A concentration. These results indicated that 40% CSE was the optimal concentration for the subsequent experiments.

### 3.2 The selection of IL-1 β overexpression and knockout vector

To test the effects of designed vectors on IL-1β expression, we transfected the vectors for IL-1β overexpression and knockdown, with the related empty vector and untransfected cells as controls. We selected three IL-1β knockout gene target vectors, which are divided into knockout 1

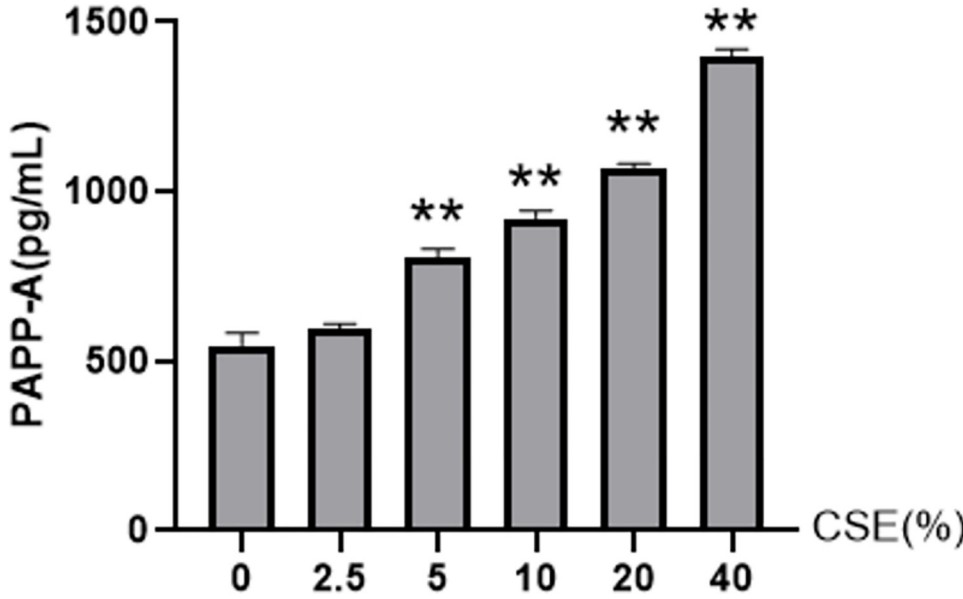

**Fig 1. Effect of CSE on PAPP-A secretions in Rat A7r5 cells.** A7r5 cells were exposed to different concentrations of CSE and incubated for 8h. PAPP-A secretion was measured in culture supernatants by ELISA. Results are represented as Mean ± S.E.M. ** P < 0.01 vs 0%CSE (n = 5). CSE:cigarette smoke extract; PAPP-A: pregnancy-associated plasma protein A.

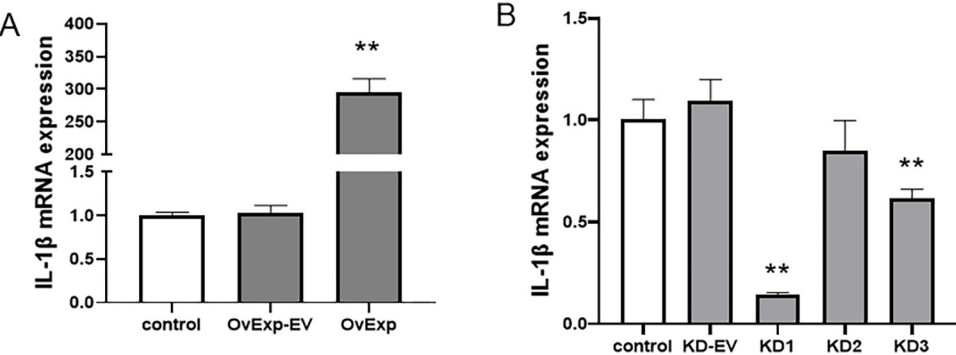

**Fig 2. The efficiency of IL-1β overexpression and knockdown.** A7r5 cells were transiently transfected with the vectors for IL-1β overexpression (A) and knockdown (B), with the related empty vector and untransfected cells as control. Cells were harvested and cell lysates were analyzed by qPCR to detect the efficiency of vectors. Results are represented as Mean ± S.E.M. ** $P < 0.01$ vs control (n = 5). **OvExp: overexpression; OvExp-EV: overexpression empty vectors; KD-EV: knockdown empty vector; KD: knockdown**.

(KD1), knockout 2 (KD2) and knockout 3 (KD3). As shown in Fig 2A, our findings exhibited no significant difference of IL-1β mRNA expression among overexpression empty vector group, knockdown empty vector group and control group ($P > 0.05$). As expected, the expression level was notably higher in group of IL-1β overexpression ($P < 0.01$). The IL-1β mRNA expression level was decreased more in KD1 and KD3 than in control ($P < 0.01$), but not KD2 ($P > 0.05$) in Fig 2B. Moreover, KD1 group has the largest reduction among the three gene knockout groups, (P<0.01). According to the results of KD1and KD2 and KD3 to target gene, cells transfected by KD1 were selected as IL-1 β knockout effector cells.

### 3.3 Knockdown of IL-1β inhibits CSE-induced apoptosis

In model group, the apoptosis rate and Bax protein expression level were significantly increased ($P < 0.01$), whereas the Bcl-2 protein expression level was significantly decreased ($P < 0.01$) when compared to control group (Fig 3A–3D). In overexpression and knockdown of IL-1β group, the apoptosis rate and Bax protein expression showed significantly respectively increased and decreased compared to model group. In contrast, the expression level of Bcl-2 protein was respectively reduced and increased on overexpression and knockdown of IL-1β group (Fig 3A–3D; P < 0.01). However, these indicators showed no significant change in the overexpression or knockdown empty vector group ($P > 0.05$). These results suggested that IL-1β knockdown markedly reduced the CSE-induced apoptosis.

### 3.4 IL-1β upregulated PAPP-A mRNA expression

The results showed that the level of PAPP-A mRNA expression in model group as well as overexpression and knockdown empty vector groups were significantly higher than control group ($P < 0.01$; Fig 4A and 4B). The IL-1β and PAPP-A mRNA expression levels were significantly higher in IL-1β overexpression group and lower in IL-1β knockdown group ($P < 0.01$). These findings indicated that CSE administration with IL-1β overexpression increased the expression levels of PAPP-A.

### 3.5 IL-1β upregulated the expression of apoptosis-inducing cytokines

The expression level of TNF-α, IL-6, and IL-8 proteins which could exacerbate apoptosis [19] was evaluated. The protein expression (TNF-α, IL-6, and IL-8) were higher in model group

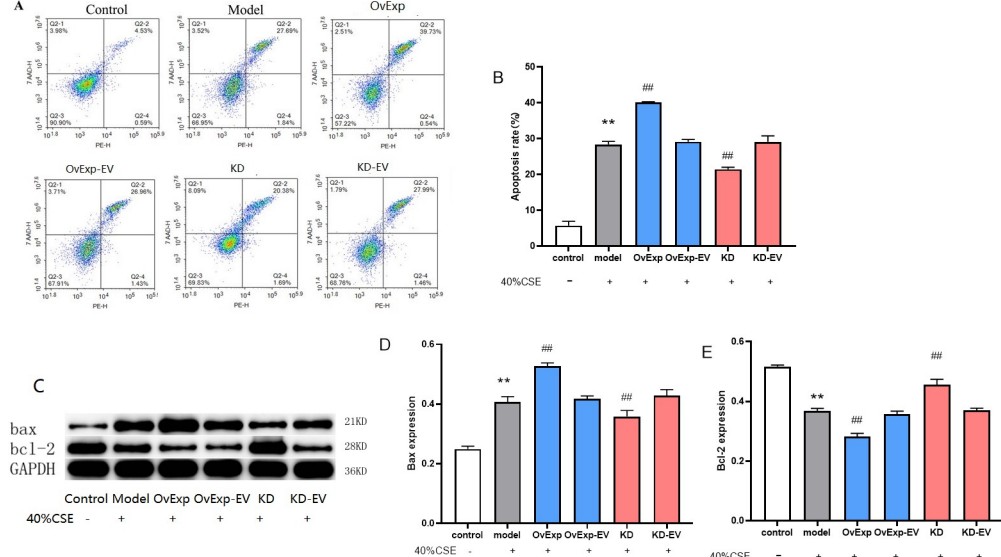

**Fig 3. The apoptosis rate and apoptosis-related protein expression levels.** The apoptosis rate was detected by flow cytometry. The increased apoptosis rate was observed in model group, which was reduced after IL-1β knockdown (A, B). The expression level of Bax protein was higher in model group and vice versa for Bcl-2, which were revered considerably by IL-1β knockdown (C-E). Results are represented as Mean ± S.E.M. $^{**}$ $P < 0.01$ vs control; $^{##}$ $P < 0.01$ vs model (n = 5).OvExp: overexpression; OvExp-EV: overexpression empty vectors; KD: knockdown; KD-EV: knockdown empty vector.

than those in control group ($P < 0.01$). No significant differences were observed among model group, overexpression empty vector group and knockdown group ($P > 0.05$). However, protein expression (TNF-α, IL-6, and IL-8) were significantly reduced in IL-1β knockdown group ($P < 0.05$ or $P < 0.01$), as shown in Fig 5A–5D. These findings indicated that the IL-1β upregulates the apoptosis-inducing cytokines, exacerbating the clinical outcomes. IL-1β knockdown significantly downregulated TNF-α, IL-6, and IL-8, which implied IL-1β cytokine could be a pharmacological target to prevent atherosclerosis induced by cigarette smoke.

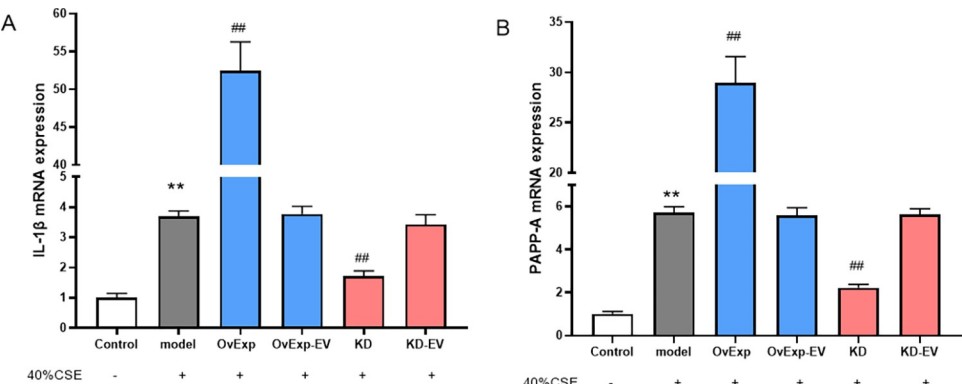

**Fig 4. IL-1β affected the expression of PAPP-A mRNA.** The increased IL-1β mRNA expression level was observed in model group, which was significantly reduced in IL-1β knockdown group (A). The PAPP-A mRNA expression level was observed in model group, which was significantly downregulated by the IL-1β knockdown (A) Results are represented as Mean ± S.E.M. $^{**}$ $P < 0.01$ vs control; $^{##}$ $P < 0.01$ vs model(n = 5).PAPP-A: pregnancy-associated plasma protein; OvExp: overexpression; OvExp-EV: overexpression empty vectors; KD: knockdown; KD-EV: knockdown empty vector.

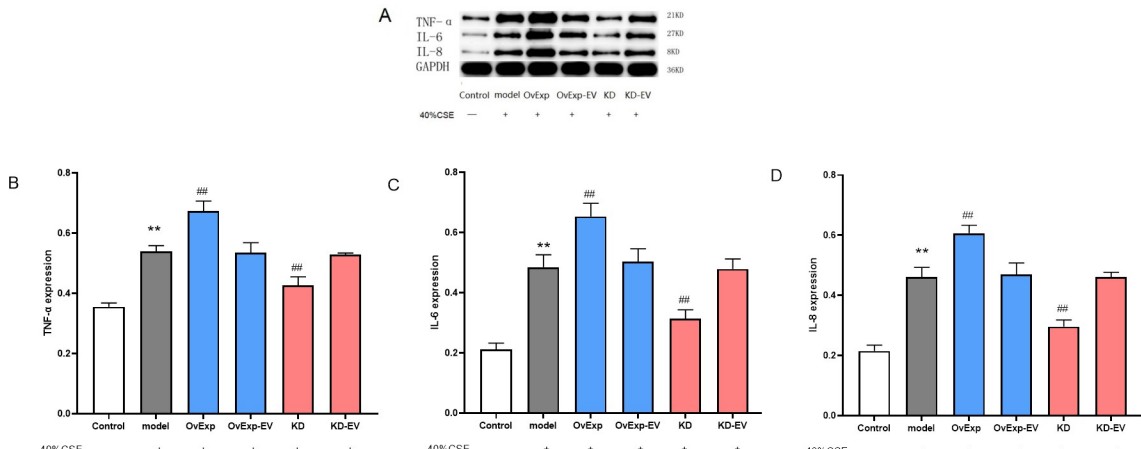

**Fig 5. IL-1β regulated the expression of TNF-α, IL-6 and IL-8.** The expression levels of TNF-α, IL-6 and IL-8 were significantly increased after CSE administration, which were downregulated upon IL-1β knockdown. Representative western blot image of the qualitative expression of TNF-α, IL-6 and IL-8 (A) The quantitative expression of TNF-α (B), IL-6 (C) and IL-8 (D) were calculated with GAPDH as an internal control. Results are represented as Mean ± S.E.M. ** $P < 0.01$ vs control; # $P < 0.05$, ## $P < 0.01$ vs model (n = 5). OvExp: overexpression; OvExp-EV: overexpression empty vectors; KD: knockdown; KD-EV: knockdown empty vector.

## 4. Discussion

In this study, cells treated with CSE in model group exhibited significantly elevated expression of IL-1β mRNA and the secretion of PAPP-A, increased expression of apoptosis related protein Bax but reduced expression of Bcl-2. In addition, the cytokines including TNF-α, IL-6 and IL-8 were also significantly increased, indicating enhanced apoptosis and inflammation. IL-1β knockdown significantly decreased cell apoptosis via regulating the expression of these proteins. Our study provided the evidence that IL-1β involves in CSE-induced PAPP-A expression and apoptosis in vascular smooth muscle cells, which might be considered as a potential clinical target for preventing of cigarette smoking-induced cardiovascular diseases.

Cigarette smoke is the leading cause of atherosclerosis [7, 20]. CSE significantly increased pro-protein convertase subtilisin/kexin type 9 (PCSK9) expression and inhibited LDLR expression in a time- and concentration-dependent manner [20]. It means that smoking can cause atherosclerosis through dyslipidemia. The dyslipidemia and the activation of inflammatory factors are the two theoretical bases of atherosclerosis. The leading cause of the cardiovascular disease, including myocardial infarction (MI), heart failure, stroke, and claudication, is atherosclerosis. [21]. Studies reported that IL-1β plays multiple roles in the interaction among inflammatory cells, smooth muscle cells, vascular endothelial cells, and extracellular matrix within the atheromatous plaque in all stages of atherosclerosis [11, 22]. IL-1β induces an inflammatory response in endothelial cells, as reflected by increased expressions of adhesion factors and chemokines, and promotes the accumulation of inflammatory cells in blood vessels and their invasion into the local intima of blood vessels, which often happens at the initiation of atherosclerosis [23].

IL-1β is a highly potent inflammatory cytokine. In this study, CSE administration altered the expression level of IL-1β in rat arterial smooth muscle cells. CSE significantly increased the mRNA level of IL-1β and protein levels of TNF-α, IL-6, and IL-8 in smooth muscle cells, which was enhanced after IL-1β overexpression and reduced after IL-1β knockdown. Previous studies indicated that IL-1β plays a vital role in the pathophysiology of cardiovascular diseases [9, 24], indicating that decrease IL-1β expression in smooth muscle cells decreases the production of atherosclerotic inflammatory factors and might prevent the progression of atherosclerosis. In contrast, increase IL-1β expression might accelerate the production of atherosclerotic

inflammatory factors and the progression of atherosclerosis. Therefore, it is critical to understand the underlying inflammatory mechanism of CSE administration and the role of IL-1β in smoking-induced atherosclerotic plaque rupture.

PAPP-A is highly expressed in eroded and ruptured atherosclerotic plaques rich in activated macrophages and smooth muscle cells and can accelerate cell lipid accumulation [25], suggesting it might be considered as one of the inflammatory mediators that predict vulnerable arterial plaques [12]. In addition, PAPP-A is related to inflammation and is regulated by inflammatory factors. Here we demonstrated that the concentration of CSE promotes the expression of the PAPP-A protein in arterial smooth muscle cells, indicating that CSE might contribute to arterial plaque rupture and arterial plaque instability. In addition, CSE-induced PPAP-A expression in a dose-dependent manner. While IL-1β overexpression or knockdown bidirectionally regulates the expression of PAPP-A. These results indicated that IL-1β might regulate PAPP-A expression in vivo.

Apoptosis and inflammation play important roles in cardiovascular disease [26]. The loss of structural integrity of mitochondria and mitochondrial dysfunction could cause extensive cellular damage and apoptosis [27]. Previous studies showed that mitochondrial proteins Bax and Bcl-2 accelerated and blocked the activated apoptotic cascade, respectively [28]. They play key roles in regulating the effect of mitochondrial membrane permeability and mitochondrial function [29]. Our study found that the apoptosis rate in model group was significantly higher (39.73%) with. considerably boosted Bax and reduced Bcl-2 expression. The apoptosis rate was significantly reduced upon (20.38%) along with a reduction of Bax expression and increased Bcl-2 level, suggesting IL-1β knockdown stabled mitochondrial membrane potential.

Moreover, this process was further synchronized with the expression of cellular inflammatory factors including TNF-α, IL-1β, IL-6, and IL-8, which increases the apoptosis rate [19]. Our findings exhibited that decreased TNF-α, IL-1β, IL-6, and IL-8 expressions were observed upon IL-1β knockdown, indicating the depreciation of the severity of apoptosis. IL-1β knockdown inhibited CSE-mediated apoptosis and down-regulated PAPP-A expression in smooth muscle cells, reflecting anti-apoptotic and anti-inflammatory effects. The overall cellular compensatory effects were achieved by IL-1β knockdown and inhibition of TNF-α, IL-6, and IL-8 expression.

We supposed that IL-1β knockdown could prevent smoking-induced thinning of the fibrous cell layer in fibrous cap of atherosclerotic plaques and minimize cardiovascular events related to fibrous cap rupture. Canakinumab Anti-inflammatory Thrombosis Outcome Study (CANTOS), a large clinical trial utilizing anti-inflammatory treatment with Canakinumab, a monoclonal antibody targeting IL-1β, demonstrated that inhibition of IL-1β expression decreased recurrent cardiovascular events, thus preventing coronary heart disease. Experimental and clinical data suggested that reducing inflammation without affecting lipid levels might decrease the risk of cardiovascular disease [30]. According to the obtained clinical data from CANTOS, a significantly lower rate of recurrent cardiovascular events were observed independent of lipid-level lowering [30]. Our results showed that CSE induced the high expression IL-1β and PAPP-A, which plays a crucial role in plaque formation and atherosclerosis, with increased risk of CVD. IL-1β knockdown led to a reduced level of IL-1β and PAPP-A, thus might decrease plaque formation and arteriosclerosis. Our study showed important practical significance for revealing the mechanism of smoking-induced cardiovascular injury, suggesting the possibility of further clinical application in the future.

## 5. Conclusions

CSE induces apoptosis and inflammatory response with increased PAPP-A expression in rat thoracic aortic smooth muscle cells. IL-1β is involved in CSE-induced PAPP-A expression and

apoptosis. Our study in vitro that CSE can show an atherosclerotic effect related to inflammatory mechanisms, more specifically to the IL-1β expression. IL-1β might be considered as a pharmacological target for preventing cardiovascular diseases caused by cigarette smoking.

### 5.1. Study limitation

Due to the animal feeding conditions of our laboratory, CSE-induced atherosclerotic inflammatory mechanism led by IL-1β has not been studied in animals.

## Supporting information

**S1 File.**
(ZIP)

## Acknowledgments

Thanks to Wuhan Myhalic Biotechnology Co., Ltd. for providing technical support.

## Author Contributions

**Conceptualization:** Hongfeng Jiang, Zhangqiang Guo.

**Data curation:** Hongfeng Jiang, Kun Zeng, Haiyan Tang, Hanxuan Tan, Caihua Huang.

**Formal analysis:** Hongfeng Jiang.

**Methodology:** Hongfeng Jiang, Zhangqiang Guo.

**Project administration:** Hongfeng Jiang.

**Supervision:** Kun Zeng.

**Validation:** Hongfeng Jiang, Rui Min.

**Writing – original draft:** Hongfeng Jiang.

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
