## [Decision Letter · Decision Letter 0]

27 Dec 2022

PONE-D-22-29892IL-1β knockdown inhibits cigarette smoke extract-induced inflammation and apoptosis in vascular smooth muscle cells.PLOS ONE

Dear Dr. Jiang,

Thank you for submitting your manuscript to PLOS ONE. After careful consideration, we feel that it has merit but does not fully meet PLOS ONE’s publication criteria as it currently stands. Therefore, we invite you to submit a revised version of the manuscript that addresses the points raised during the review process.

We look forward to receiving your revised manuscript.

Kind regards,

Yi Cao

Academic Editor

PLOS ONE

Journal Requirements:

Reviewers' comments:

Reviewer's Responses to Questions

**Comments to the Author**

1. Is the manuscript technically sound, and do the data support the conclusions?

Reviewer #1: Yes

2. Has the statistical analysis been performed appropriately and rigorously? 

Reviewer #1: Yes

3. Have the authors made all data underlying the findings in their manuscript fully available?

Reviewer #1: Yes

4. Is the manuscript presented in an intelligible fashion and written in standard English?

Reviewer #1: Yes

5. Review Comments to the Author

Reviewer #1: In this in vitro study, the authors investigate the role of IL-1β and other cytokines, and apoptotic mechanisms in the genesis of atherosclerosis induced by cigarette smoke in vascular smooth muscle cells. They showed that cigarette smoke has pro-inflammatory and apoptotic effects that were abolished in IL-1β knockdown cells. Thus, IL-1β cytokine is a pharmacological target to prevent atherosclerosis induced by cigarette smoke. The results are interesting and add knowledge to the area.

I have few concerns:

**No page numbers. This makes the review complicated.

Abstract

-page 1, line 20: CSE administion = administration

-page 1, line 23 – The conclusion needs a grammar review (IL-1β involves = IL-1β is involved…and other parts)

-page 1, KEYWORDS. Words in the title are already keywords.

Introduction

-Page 2 lines 11 and 12– the word “atherosclerosis” is repeated. Please, be concise.

- Globally the introduction can be reduced because there are many unneeded words, and some ideas are repetitive. For example, in lines 16 to 25; or lines 26-28… Specifically, the phrase: “IL-1β is essential for vascular inflammation” does not add any new information. The same at page 2, “Many studies support that IL-1β involves in cardiovascular disease”. Details that the authors think are important to understanding their results are better located in the discussion. I recommend revising this part of the manuscript.

- page 3, line 7: “which is a novel metalloproteinase”. I didn’t understand what “novel” means. It is new in the acknowledgment, right?

-- page 3, line 20: pro-inflammatory cytolines = cytokines

-- page 3, lines 24 to 29 and the next page until “We aimed to investigate.. are information more appropriate for the method section.

Results

-Pg 7, lines 19-20: “To investigate the effect of CSE administration on the expression level of PAPP-A, the concentrations of PAPP-A in culture medium was detected.” Is this phrase needed? Please, be concise.

-Page 8, figure 1 legend is incomplete. The authors need to show what CSE, and PAPP-A abbreviations are, so the reader can understand their results without reading the complete manuscript (the same for figures).

-- Fig 2 shows A and B independent graphics, however, the legend did not clarify which information each figure represents. What is EV? Figures should be explained without text support.

--page 9: … significantly respectively ... (?); … In contranst,

--page 10, figure 3, the phrase “The expression levels of apoptosis-related proteins were detected by Western blot” does not add information and is not appropriate for a legend. I also recommend avoiding the excessive use of the word “significantly”. It is obvious that the mathematics proved that p < 0.05, results are different.

--page 10: 2.4 IL-1β “reguglated” ? …. reduced PAPP-A “expresson”(?)

--page 10: “Then the effect of IL-1β knockdown on PAPP-A mRNA expression was determined.” = Unneeded phrase.

--page 11, fig 4: “The mRNA expression levels were determined by qPCR.” This information is already shown in the method section and does not help to understand the figure results. I also recommend revising the legend, avoiding unneeded words.

-- page 11, …significantly “recuded”(?); … might be a “potential”(?) target

--page 12: … inflammation “aossociated” (?) proteins

Discussion

--page 13: The main “companet”(?) of, …which is “an” critical factor..

--page 13: The authors attributed nicotine to the cause of the increased risk of atherosclerosis. However, they did not show any information from the literature confirming this statement. Moreover, they did not discuss other toxic compounds with more pro-inflammatory effects than nicotine. Most of them are carcinogenic too. Is there any study proving the cardiotoxic effect of nicotine? How could other cigarette compounds be responsible for the pro-inflammatory effects of cigarette smoke? If the authors think that nicotine is the most important source of the cardiotoxic effect why they did not treat the cells with this isolated compound? Reference 20 does not state that “Cigarette smoke is the leading cause of atherosclerosis (20)”.

--page 14: … mitochondrial “proteions”. ... knockdown and “inhibiton”

--page 15: … “expresson” IL-1β and PAPP-A; … and “atherosclerosi.”; … in the “furture.”

Conclusion:

-- please revise: CSE “could induce” apoptosis …(?). According to their results, the authors can confirm that “CSE induces apoptosis and …”.

-- IL-1β “is involved” in CSE-induced PAPP-A expression and apoptosis.

In this section, the authors repeated information from their results without a more conclusive statement. Indeed, they showed in vitro that cigarette smoke can show a cardiotoxic effect related to inflammatory mechanisms, more specifically to the IL-1β expression. Additionally, I do not agree with the statement that “IL-1β might be considered as a potential clinical target in the treatment of cardiovascular diseases” but a target for preventing cardiovascular diseases caused by cigarette smoking. It is a challenging question to think about.

6. PLOS authors have the option to publish the peer review history of their article (what does this mean?). If published, this will include your full peer review and any attached files.

Reviewer #1: No

---

## [Author Response · Author response to Decision Letter 0]

26 Jan 2023

Dear Editor and Reviewer #1 : 

Thank you for carefully reviewing our manuscript previously titled “IL-1β knockdown inhibits cigarette smoke extract-induced inflammation and apoptosis in vascular smooth muscle cells ” for possible publication in the PLOS ONE. We are grateful to reviewer #1 for your effort reviewing our paper and your positive feedback. The summary of our work as written by this reviewer #1 is precise. We have revised the manuscript, highlighting our revisions in red. and have attached point-by-point responses detailing how we have revised the manuscript in response to the reviewers' comments below.

Thank you for your consideration and further review of our manuscript. Please do not hesitate to contact us with any further questions or recommendations.

Yours Sincerely,

Hongfeng Jiang

---

## [Editor Report · Decision Letter 1]

1 Feb 2023

IL-1β knockdown inhibits cigarette smoke extract-induced inflammation and apoptosis in vascular smooth muscle cells.

PONE-D-22-29892R1

Dear Dr. Jiang,

We’re pleased to inform you that your manuscript has been judged scientifically suitable for publication and will be formally accepted for publication once it meets all outstanding technical requirements.

Kind regards,

Yi Cao

Academic Editor

PLOS ONE
---

## [Editor Report · Acceptance letter]

6 Feb 2023

PONE-D-22-29892R1 

IL-1β knockdown inhibits cigarette smoke extract-induced inflammation and apoptosis in vascular smooth muscle cells 

Dear Dr. Jiang:

I'm pleased to inform you that your manuscript has been deemed suitable for publication in PLOS ONE. Congratulations! Your manuscript is now with our production department. 

Kind regards, 

on behalf of

Dr. Yi Cao 

Academic Editor

PLOS ONE